# Distinctive Patterns of 5-Methylcytosine and 5-Hydroxymethylcytosine in Schizophrenia

**DOI:** 10.3390/ijms25010636

**Published:** 2024-01-04

**Authors:** Jiaxiang Xie, Yang Wang, Changcheng Ye, Xiao-Jiang Li, Li Lin

**Affiliations:** Guangdong Key Laboratory of Non-Human Primate Research, Laboratory of CNS Regeneration (Ministry of Education), Guangdong-Hongkong-Macau Institute of CNS Regeneration, Jinan University, Guangzhou 510632, China; tsegaaicoeng@hotmail.com (J.X.); wangyang@stu2022.jnu.edu.cn (Y.W.); changchengye@stu2022.jnu.edu.cn (C.Y.); xjli33@jnu.edu.cn (X.-J.L.)

**Keywords:** schizophrenia, epigenetic, DNA methylation, 5mC, 5hmC

## Abstract

Schizophrenia is a highly heritable neuropsychiatric disorder characterized by cognitive and social dysfunction. Genetic, epigenetic, and environmental factors are together implicated in the pathogenesis and development of schizophrenia. DNA methylation, 5-methycytosine (5mC) and 5-hydroxylcytosine (5hmC) have been recognized as key epigenetic elements in neurodevelopment, ageing, and neurodegenerative diseases. Recently, distinctive 5mC and 5hmC pattern and expression changes of related genes have been discovered in schizophrenia. Antipsychotic drugs that affect 5mC status can alleviate symptoms in patients with schizophrenia, suggesting a critical role for DNA methylation in the pathogenesis of schizophrenia. Further exploring the signatures of 5mC and 5hmC in schizophrenia and developing precision-targeted epigenetic drugs based on this will provide new insights into the diagnosis and treatment of schizophrenia.

## 1. Introduction

Schizophrenia is a serious and debilitating neurodevelopmental disorder that typically occurs in early adulthood. Patients experience hallucinations, delusions, disorganization, and deterioration in cognitive and social functioning, such as amotivation and social withdrawal [1]. The average lifetime prevalence of schizophrenia is just under 1% [2]. Individuals from all ages with schizophrenia have high mortality and have their life expectancy reduced by approximately 20 years compared with the general population [3]. Alterations in cortical circuits have been reported to be involved in the pathogenesis of schizophrenia. The loss of dendritic spines in pyramidal cells results in reduced excitatory activity, while, at the same time, reduced excitatory input to GABAergic interneurons results in reduced inhibition in pyramidal cells. These changes are thought to lead to an abnormal functional network [4].

The heritability of schizophrenia is as high as 80% [5]. Genetic factors, including copy number variants [6] and *de novo* mutations [7], have been identified to be closely associated with the development of schizophrenia. Interestingly, studies of monozygotic twins have revealed the heterogeneity of schizophrenia, suggesting that non-genetic factors may be involved in the pathogenesis of schizophrenia. Environmental factors can induce enduring changes in gene expression through epigenetic mechanisms in neurodevelopment processes, aging, and neurodegenerative diseases [8]. DNA methylation, a critical epigenetic modification, contributes to alter gene expression without affecting the underlying genomic sequences; 5-methycytosine (5mC) and 5-hydroxylcytosine (5hmC) are two major forms of DNA methylation in mammals. DNA methyltransferases (DNMTs) transfer a methyl group to the 5-position of cytosine to form 5mC; the ten-eleven translocation (TET) dioxygenases catalyze the 5mC to 5hmC, 5-formylcytosine (5fC), and 5-carboxylcytosine (5caC) on DNA.

Altering the expression or activity of DNA methylation-related enzymes produce exacerbated phenotypes or therapeutic effects in different animal models. Cathrin Bayer et al. found that the depletion of DNMT1 increased the survival of mutant Huntington-transfected cells and accelerated perinuclear Huntington aggregation, suggesting that DNMT1-dependent degradation pathways mediate mutant Huntington-induced cytotoxicity [9]. Growing evidence suggests that epigenetic responses to environmental stimuli and plays an important role in schizophrenia [10]. By mapping thousands of 5mC sites in the prefrontal cortex of schizophrenia patients and healthy controls, differentially methylated regions (DMRs) at genes related to development and neurodifferentiation have been identified [11]. Disruptions of genome-wide 5mC have been observed in peripheral blood mononuclear cells (PBMCs) of schizophrenia patients, particularly in those experiencing their first episode of disease [12]. Despite inconsistent results from genome-wide association studies (GWAS), 108 replicable genomic loci have been identified in 2014 [6].

Currently, antipsychotic drugs that can improve positive symptoms are clinically used to treat schizophrenia [3]. Drug treatment has been shown to alter the DNA methylation status. Therefore, the development of epigenetic treatments targeting DNA methylation is helpful for schizophrenia. Further understanding of the cellular and molecular mechanisms of schizophrenia is necessary to develop new therapeutic strategies. In this review, we summarized the potential roles of 5mC, 5hmC, and their related enzymes in the pathological process of schizophrenia. Additionally, we discuss the epigenetic-based therapeutic approaches to alleviate the symptoms of schizophrenia.

## 2. Altered 5mC Pattern in Schizophrenia

It is known that 5mC is the primary product of DNA methylation, which can influence various processes such as gene activity, individual development, and cancer progression [13,14]. DNMT1, DNMT3A, and DNMT3B possess catalytic activity and covalently add a methyl group from S-adenosylmethionine (SAM) onto the fifth carbon of the cytosine (C) pyrimidine ring, resulting in the formation of 5mC. While DNMT2 and DNMT3L contain conserved sequences of DNMT enzymes, they lack catalytic activity [15,16]. During late embryonic development and after maturation, genomic methylation patterns are established, and DNMT1 is responsible for maintaining the genomic methylation status. The DNMT1 is approximately one hundred times more active than DNMT3A and DNMT3B [17].

Abnormalities of 5mC have been widely reported in neurodegenerative diseases, such as Parkinson’s disease (PD) and Huntington’s disease (HD) [18,19,20]. Psychiatric disorders are characterized by impairment in cognition, emotion regulation, or behavior, and typically include bipolar disorder (BD) and schizophrenia. The pathogenesis of BD involves mitochondrial dysfunction, DNA oxidative damage, and DNA methylation changes. BD patients and their relatives exhibit genome-wide decreases in 5mC levels. After lithium treatment, 5mC levels remained unchanged in BD patients, whereas the levels were increased in their relatives, providing evidence of the significance of 5mC modification in BD pathology [21,22,23].

The overexpression of DNMT1 was considered to be one of the etiological factors of schizophrenia. Transcriptional dysregulation has been observed in embryonic stem cells overexpressing DNMT1. Approximately 50% of these genes have been implicated in schizophrenia, showing dysregulation independent of DNA methylation [24,25]. A novel GWAS in patients with schizophrenia identified schizophrenia-associated differential 5mC at 242 sites, of which mitotic arrest deficient 1-like 1 (MAD1L1) was robustly differentially methylated [26].

Alongside GWAS, a growing number of studies aim to identify the 5mC of candidate genes in patients with schizophrenia [27]. Specific 5mC differences have also been found in the gray and white matter of individuals with schizophrenia. These DMRs were identified within or near the Kruppel-like factor 9 (KLF9), sideroflexin 1 (SFXN1), Sprouty-related EVH1 domain-containing 2 (SPRED2), and AlS2 C-terminal-like (ALS2CL) genes [28]. Other key genes with dynamic 5mC were also involved in the pathogenesis of schizophrenia (Table 1).

Reelin protein is a signaling molecule that plays a crucial role in the nervous system. In humans, the dysregulation of Reelin protein is associated with various neurological disorders, including BD, autism, depression, and schizophrenia [39,40]. It is encoded by the RELN gene and was originally discovered in recessive mutant mice. The abnormal expression of this protein can lead to manifestations such as gait disturbances, ataxia, and tremors [39]. Reelin protein regulates the adhesion and migration of nerve cells by activating receptors on the cell surface, thereby promoting the proper positioning and layering of cerebral cortex neurons [39]. In humans, the dysregulation of Reelin protein is associated with various neurological disorders, including schizophrenia, BD, autism, depression, and more [39,40]. Reelin was found to be altered in various animal models of schizophrenia, suggesting its key role in the pathogenesis of schizophrenia [41]. Reelin levels were significantly reduced in schizophrenia patients, with specific gender differences [42]. The reduction of Reelin in schizophrenia was attributed to a significant increase in 5mC levels at the RELN promoter, with unmethylated Reelin expressed 25-fold higher than methylated Reelin [29,30]. Therefore, 5mC is an important factor in regulating RELN expression, and exploring the precise mechanism of the methylation of RELN may help elucidate the pathogenesis of schizophrenia.

Brain-derived neurotrophic factor (BDNF) regulates neurogenesis and synaptic plasticity as a biomarker for neuropsychiatric disorders [31]. such as bipolar disorder and major depressive disorder, and alterations in BDNF levels, the imbalance between pro-BDNF and m-BDNF, and defects in the BDNF signaling pathway have been observed [43]. When the BDNF Val66Met was knocked into mice, BDNF expression was significantly decreased, resulting in reduced neurogenesis and structural abnormalities in the hippocampus [44,45]. In patients with chronic schizophrenia, BDNF levels are closely related to memory, attention, information-processing speed, and cognitive impairment [46]. The 5mC of the BDNF gene could be regulated by environmental conditions and was believed to be involved in the pathogenesis of schizophrenia. Using bisulfite sequencing, no differences were found in BDNF 5mC status between schizophrenia patients and healthy controls, but there was a correlation between disease progression and 5mC [47]. An analysis of postmortem brains from schizophrenia patients revealed elevated levels of 5mC and 5hmC in the BDNF-regulatory regions in the frontal cortex and hippocampus, including the promoter regions, suggesting a strong association between BDNF methylation and schizophrenia [31]. Furthermore, schizophrenia patients with the same genotype and alleles exhibited different symptoms, which may be related to the differences in BDNF expression levels. Additionally, BDNF levels may differ in schizophrenia patients treated with different drugs [48].

The specific mechanisms underlying schizophrenia is still completely unclear, but classical theories believe that serotonin, glutamate, and dopamine networks are closely associated with the onset of schizophrenia [49]. Glutamate decarboxylase 1 (GAD1) is an important factor in the glutamate metabolism network; it plays a crucial role in conditions such as attention deficit/hyperactivity disorder (ADHD), heroin addiction, early infantile epilepsy, and developmental delay [50,51,52]. In patients with schizophrenia, 5mC levels in the promoter region of GAD1 were reduced, resulting in a significant increase in the overall expression of GAD1 [37,53]. The COMT gene encodes catechol-O-methyltransferase and is involved in the regulation of the dopamine metabolism network, it is involved in treatment-resistant depression, borderline personality disorder, post-traumatic stress disorder, and other mental disorders [54,55,56]. In patients with schizophrenia, the 5mC levels of the COMT gene are elevated, resulting in a significant increase in mRNA expression. The dysregulation of COMT expression has been demonstrated in the pathophysiology of schizophrenia [38,57,58] Dopamine transporter (DAT) is encoded by the Solute Carrier Family 6 Member 3 (SLC6A3) gene and is located on the neuronal cell membrane. It is responsible for the reuptake of dopamine from the synaptic cleft into neurons, thereby regulating the concentration and duration of the dopamine action in neurons. Therefore, it plays a role in emotional control, reward regulation, and other processes [59,60]. Moreover, 5mC levels in the DAT 5′-UTR of PD patients were different from healthy controls, and this methylation level may be related to different stages of the disease [61]. Cognitive impairment is one of the symptoms in schizophrenia patients. DAT availability in the striatum has been showed to be positively correlated with cognitive function in schizophrenia patients. Reduced DAT availability was primarily associated with increased 5mC levels [32,33]. These findings indicated that DNA methylation regulates key genes in the dopamine-network-involved pathological progress of schizophrenia.

Dystrobrevin-binding protein 1 (DTNBP1) is also highly associated with schizophrenia [34]. DTNBP1 is considered to be one of the key genes in the glutamatergic system; it also plays a role in depression and temporal lobe epilepsy [62,63]. It can interact with BDNF to regulate activity-dependent BDNF secretion, neural development, and inhibitory circuit function. Abnormalities in the DTNBP1 pathway may lead to schizophrenia [64,65]. In rodents, the reduced expression of DTNBP1 resulted in abnormal neuronal growth and morphology. Additionally, DTNBP1 plays a role in schizophrenia by regulating neurotransmission through the modulation of N-methyl-D-aspartate receptors (NMDAR) and dopamine D2 receptors (DRD2) [34]. It has been reported that female schizophrenia patients have higher 5mC levels at the upstream CpG sites of DTNBP1. This result was consistent with the observed downregulation of DTNBP1 mRNA levels in schizophrenia patients. Furthermore, investigations of saliva and postmortem brain samples from schizophrenia patients have revealed increased levels of 5mC in the DTNBP1 promoter, suggesting the potential role for 5mC on DTNBP1 in schizophrenia [35,36]. To sum up, the above related 5mC changes are summarized in Figure 1.

## 3. Altered 5hmC Pattern in Schizophrenia

In as early as 1952, 5hmC was discovered in bacteriophage DNA [66] and was first detected in mammalian genomes in 1972 [67]. As of 2009, the TET family has been identified to convert 5mC to 5hmC both in vivo and in vitro [68,69]. The TET protein family includes three members, TET1, TET2, and TET3, which have a conserved double-stranded β-helix domain (DSBH) at their C-terminus, which contains α-ketoglutarate (α-KG)- and Fe(II)-binding sites. Additionally, there is a cysteine-rich domain (CD) that contains a high concentration of cysteine residues. Together, these two domains form the core catalytic domain, which mediates the oxidation reaction of 5mC to 5hmC [70].

The distribution of 5hmC in animal tissues is widespread but variable, displaying certain tissue specificity [71,72,73]. It has been found, through 5hmC profiling in mammals, that 5hmC is mainly enriched in the central nervous system (CNS), with lower levels in peripheral tissues, which is in sharp contrast to the global distribution of 5mC [73,74]. In the brain, 5hmC exhibits region-specific distribution, with a unique distribution pattern in the cerebellum [75,76]. This suggests that 5hmC may play an important role in tissue-specific functions. Furthermore, 5hmC levels increase during neuronal differentiation and are associated with genes critical for neuronal function during neurodevelopment [77]. In addition, 5hmC is involved in processes of cell development and differentiation, the regulation of chromatin structure, and bone marrow regeneration [78,79,80]. In the process of DNA methylation regulation, the epigenetic plasticity of the CNS affects the expression of neuroactivity-dependent genes, organism learning, and memory [81,82].

It is well-known that 5mC is considered a stable heritable modification and represents a static process in epigenetic regulation. In fact, the dynamic balance between 5mC and 5hmC is an important condition for organism homeostasis, and imbalances in their ratio are implicated in the pathogenesis of many neurodegenerative diseases [78,79,83,84]. Abnormalities in 5hmC have been reported to be widespread in patients and animal models of AD and PD. At the same time, it was found that dynamic changes in 5hmC were positively correlated with aging [76,80].

Genome-wide DNA methylation changes may lead to genomic instability, and DNA methylation changes at promoter regions typically affect gene transcription. In schizophrenia, genome-wide 5hmC levels were elevated in male patients and reduced in female patients compared with healthy individuals [10], suggesting a potential association between the dysregulation of 5hmC levels and the development of schizophrenia. Many candidate genes in the peripheral blood or postmortem brain from schizophrenia patients have been identified with the differential methylation status associated with schizophrenia [11,85], including nitric oxide synthase 1 (NOS1), AKT serine/threonine kinase 1 (AKT1), DTNBP1, DNMT1, protein phosphatase 3 catalytic subunit gamma (PPP3CC), glutamic acid decarboxylase 67 (GAD67), and sex-determining region Y-box containing gene 10 (SOX10) [86,87]. These genes exhibit varying degrees of changes in 5mC or 5hmC (Figure 1 and Table 2).

In the DNA demethylation pathway, apolipoprotein B mRNA-editing enzymes (APOBEC) are an important lipid carrier protein and participate in the conversion of 5hmC to 5fC and 5caC. TET1 expression was increased, while APOBEC3A and APOBEC3C expression was decreased in the frontal cortex of schizophrenia patients, suggesting that the disruption of the DNA methylation and demethylation pathways may contribute to elevated 5hmC levels in schizophrenia patients [87]. In addition, it has also been found that the increase in 5hmC occurring in the promoter region of the BDNF gene was associated with a reduced expression of BDNF in the frontal cortex of patients with schizophrenia [89]. Increased levels of 5mC and 5hmC in the BDNF promoter reduced the binding of GADD45b protein in the associated chromatin regulatory region and further reduced transcription levels in schizophrenia patients [89]. Interestingly, our unpublished data from the 5hmC profiling of different brain regions in rhesus monkey revealed that cerebellum-specific differentially hydroxymethylated regions (DhMRs) were concentrated in pathways associated with schizophrenia, including GRIN2C and PAX6.

The dysregulation of methylation levels in schizophrenia-related genes plays a crucial role in the development of schizophrenia. However, research on 5hmC changes in schizophrenia remains limited. Further studies are needed to elucidate the detailed mechanisms by which 5hmC contributes to schizophrenia.

## 4. Therapeutic Potential for Schizophrenia Based on DNA Methylation Changes

Schizophrenia is a chronic mental disorder for which there are still no clear pathophysiological mechanisms, precise biomarkers, and approved specialized therapeutic drugs. Currently, antipsychotic medications are commonly used in the treatment of schizophrenia that could effectively alleviate positive symptoms, including chlorpromazine (CPZ), risperidone (RIS), olanzapine (OLZ), and aripiprazole (ARI).

CPZ is considered a first-line treatment for treatment-resistant schizophrenia. Significant changes in 5mC at 29,134 CpG sites were reported in the white blood cells of patients after one year of CPZ treatment. Among these sites, 13,052 showed increased 5mC levels, while 16,082 showed decreased 5mC levels. Approximately 60% of the observed decreases in 5mC occurred in the promoter regions of genes, including schizophrenia susceptibility genes such as GAD1 and glutamate metabotropic receptor 7 (GRM7). In fact, patients with schizophrenia have increased 5mC in the GAD1 promoter region, resulting in decreased expression. CPZ treatment successfully reversed GAD1 expression [37,90,91]. OLZ is a second-generation antipsychotic drug widely used to treat schizophrenia and BD. It affects various neurotransmitter systems in the CNS, including the dopamine and norepinephrine systems [92]. OLZ treatment in rats affected the 5mC levels of several genes associated with schizophrenia, including dopamine D1 receptor (DRD1), dopamine D2 receptor (DRD2), dopamine D5 receptor (DRD5), and others. The dopamine system is considered to be closely related to schizophrenia. OLZ treatment increased 5mC in the promoter region of the above genes, reduced dopamine activity, and improved schizophrenia symptoms [88,93]. Interleukin 6 (IL-6) is a cytokine involved in immune responses and is considered a potential peripheral biomarker for schizophrenia. Patients with schizophrenia showed lower levels of 5mC in the IL-6 promoter region compared with healthy controls. This hypomethylation status in the promoter was reversed upon treatment with different antipsychotic drugs [94,95].

Combination therapy involves the simultaneous use of two or more medications and is a common approach in clinical psychiatry [96]. However, since schizophrenia involves multiple genetic and epigenetic loci, treatment responses and side effects vary among individuals, emphasizing the need to find complete and effective treatments [97]. Epigenetic profiling offers a promise for the development of diagnostic and therapeutic biomarkers for schizophrenia. The integrated analyses of DNA methylation and gene expression to identify aberrant DNA methylation will advance this goal.

## 5. Conclusions and Perspective

Schizophrenia is a complex disorder which involves physiological, psychological, and environmental factors [1]. It has a clinical heterogeneity and polygenic effects in disease development. Analyses of genetic risk loci showed that many sites are located in the regulatory regions such as promoters and introns, which are closely associated with gene expression. This implicated that epigenetic-modification-mediated gene regulation plays a key role in the pathogenesis of schizophrenia. Indeed, there is growing evidence that DNA methylation, specially 5mC and 5hmC, is linked to schizophrenia. Both genome-wide 5mC patterns and candidate gene 5mC and 5hmC patterns suggest a regulatory role for DNA methylation alterations in the development of schizophrenia.

Schizophrenia is divided into several subtypes based on clinical features. The diagnostic criteria in the DSM-5 divide schizophrenia into five subtypes. However, as the number of detected cases increases, these five subtypes are no longer sufficient to comprehensively cover the schizophrenia spectrum. Therefore, a further refined subdivision of schizophrenia subtypes is needed, as DNA methylation can also differ in different types of schizophrenia [98]. Dynamic DNA methylation was also associated with several schizophrenia types outside the DSM-5 classification. Early-onset schizophrenia refers to the onset of schizophrenia before the age of 13 and progresses with age. Hypermethylation of CpG site cg10392614 on chromosome 2 was closely associated with early-onset schizophrenia [99,100]. Paranoid schizophrenia patients exhibited significant changes in the partial methylation pattern of long interspersed nuclear elements-1 (LINE-1) [101]. Deficit-type schizophrenia is characterized by primary and enduring negative symptoms. Increased C-X-C motif chemokine ligand 1 (CXCL1) expression and hypomethylation have been observed in patients, suggesting that DNA methylation is one of the regulatory factors controlling CXCL1 expression at the transcriptional level [102]. In recent years, there has been increasing recognition of the importance of the gut microbiota and its impact on the CNS. The gut microbiota may potentially influence schizophrenia by modulating key metabolic pathways and epigenetic modifications. However, there is currently a lack of direct evidence in this regard, and further research is needed to explore this relationship in the future [103].

The clinical heterogeneity of schizophrenia complicates its diagnosis. To better understand the pathophysiology of the disorder, further investigations in postmortem and peripheral tissues are needed. Comprehensive multiomics analyses will provide insights into the interaction between epigenetic modifications and gene expression and reveal the precise mechanisms of schizophrenia. On this basis, molecular biomarker screening and therapeutic drug intervention will lead to new treatment directions.

## Figures and Tables

**Figure 1 ijms-25-00636-f001:**
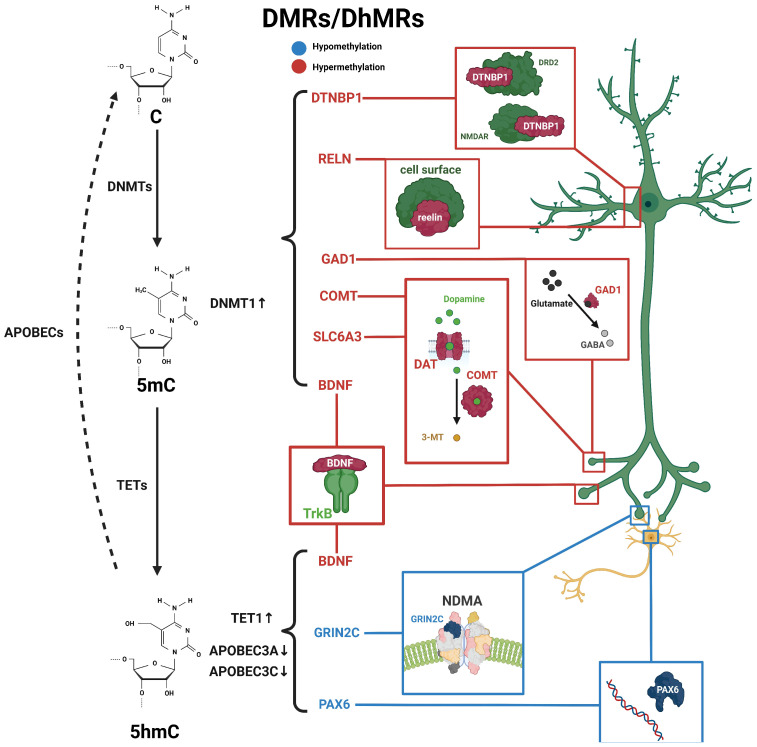
Schematic diagram of dynamic 5mC and 5hmC in neurons of schizophrenia patients. 5mC is catalyzed by DNMTs from cytosine and is then hydroxylated to 5hmC by TET enzymes, and further conversed into cytosine through the participation of apolipoprotein B mRNA-editing enzymes (APOBEC). Alterations in 5mC and 5hmC patterns have been identified in patients with schizophrenia. BDNF processes both 5mC and 5hmC dynamics. These distinctive patterns of 5mC and 5hmC are implicated in the pathogenesis and development of schizophrenia. Blue, hypomethylation; red, hypermethylation.

**Table 1 ijms-25-00636-t001:** Association of 5mC with schizophrenia.

DMRs	Tissues	Expression	5mC	Phenotypes	References
KLF9	Human cortical grey and white matter	↓	↑	rs11142387 near the KLF9 was significantly associated with psychiatric disease and poor memory function.	[28]
SFXN1	Human cortical grey and white matter	↓	↑	The loss of SPRED2 leads to defective glycine and purine synthesis.	[28]
SPRED2	Human cortical grey and white matter	↓	↑	The loss of SPRED2 leads to a phenotype resembling recessive Noonan syndrome.	[28]
ALS2CL	Human cortical grey and white matter	↑	↓	Mutations in ALS2CL may contribute to the development of schizophrenia.	[28]
RELN	Human peripheral blood	↓	↑	Single-allele and biallelic mutations in RELN can lead to neurodevelopmental disorders. The dysregulation of RELN expression has been observed in patients with schizophrenia and bipolar disorder.	[29,30]
BDNF	Human peripheral blood	↓	↑	BDNF activates the tyrosine kinase receptor B (TrkB), triggering various downstream signaling pathways. In patients with schizophrenia, there are alterations in BDNF signaling transduction.	[31]
SLC6A3	Isohelix swab pack	↓	↑	SLC6A3 is associated with several neurological and psychiatric disorders, including ADHD, autism, cognitive impairments, movement disorders, and schizophrenia.	[32,33]
DTNBP1	Human brain	↓	↑	The aberrant expression of DTNBP1B is associated with cognitive deficits in schizophrenia.	[34,35,36]
GAD1	Human	↓	↑	The GAD1-knockout mouse model exhibits impairments in spatial memory and working memory. It shows reduced locomotor activity in new environments and a decreased preference for novel stimuli.	[37]
COMT	Human peripheral blood	↑	↑	The deletion of the COMT gene can lead to a range of complex complications, with psychiatric symptoms manifesting as schizophrenia and other mental disorders.	[38]

↑: upregulation; ↓: downregulation.

**Table 2 ijms-25-00636-t002:** Association of 5hmC with schizophrenia.

DhMRs	Tissues	Expression	5hmC	Phenotypes	References
GABRB2	Human, peripheral white blood cells	↓	↑	Gabrb2-knockout mice exhibit anxiety-like and depression-like behavioral changes, as well as alterations in social behavior, learning, and memory abilities.	[85]
GAD67	Human, parietal cortex	↓	↑	GAD67-knockout mice exhibit emotional and auditory abnormalities, as well as anxiety-like behavior.	[87,88]
APOBEC3A/C	Human, parietal cortex and prefrontal cortex	↓	↑	The deletion of APOBEC3A has been associated with an increased susceptibility to early-onset breast cancer.	[87]
GADD45b		↑	↑	The knockdown of Gadd45b in the amygdala of neonatal rats leads to changes in social behavior during adolescence and a decrease in the expression of several genes associated with psychiatric disorders, including MeCP2, Reelin, and BDNF.	[89]
BDNF IX		↓	↑	BDNF knockout mice exhibit chronic liver disease, specifically non-alcoholic fatty liver disease (NAFLD).	[89]
GRIN2C	Monkey, cerebellum	↓	-	The knockdown of PAX6 in differentiating human limbal epithelial cells leads to a decrease in the expression of FABP5 and DSG1 proteins.	unpublished data
PAX6		↑	-		unpublished data

↑: upregulation; ↓: downregulation.

## Data Availability

Not applicable.

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
