# Peer review of "Distinctive Patterns of 5-Methylcytosine and 5-Hydroxymethylcytosine in Schizophrenia"

_ijms, 2024, doi:10.3390/ijms25010636_

Round 1

Reviewer 1 Report

Comments and Suggestions for Authors

In the article entitled "Distinctive patterns of 5-methylcytosine and 5-hydroxymethyl-cytosine in schizophrenia," the authors reviewed the role of DNA methylation and hydroxymethylation in the pathogenesis of schizophrenia. It highlights the discovery of unique patterns and expression changes of 5-methylcytosine (5mC) and 5-hydroxymethylcytosine (5hMC) in schizophrenia. The work suggests that antipsychotic drugs that affect DNA methylation status can alleviate symptoms in patients with schizophrenia, indicating the critical role of DNA methylation in the development of the disorder. The authors propose further exploration of 5mC and 5hMc signatures in schizophrenia to develop precisely targeted epigenetic drugs for diagnosis and treatment.

The article is very comprehensive and up to date on the topic of methylation in schizophrenia.

Author Response

We thank the reviewer for the affirmation of the manuscript.

Reviewer 2 Report

Comments and Suggestions for Authors

In this review, Jiaxiang Xie and colleagues analyzed distinctive patterns of 5-methylcytosine and 5-hy-droxymethylcytosine in schizophrenia. This analysis is justified by research indicating that the alteration in epigenetic pathway was associated with clinical features and brain dysfunctions in schizophrenia. Although the rationale seems quite reasonable, the manuscript has many serious shortcomings. I suggest the Authors clarify or provide more details regarding the following major and minor points.

Major concerns:

1. It is well known that many neuropsychiatric diseases, and others, are associated with epigenetic changes. Many post-mortem tests showed that that the level of DNMT and DNA methylation is higher in patients with schizophrenia than in healthy people. The cases of epigenetic modifications of the promoters of genes encoding COMT, Reelin and others described in this manuscript are well documented. They are not specific to schizophrenia and are also observed in other neuropsychiatric diseases. Therefore, the authors should clearly state the purpose of preparing this review.

2. Did the authors intend to present a systematic review? Because if so, they should use the methodology applicable to reviews of this type.

3. It should be emphasized what is new in the approach presented here. What's new in the analysis of generally known data from the scientific literature on epigenetic changes described in schizophrenia? Any new conclusions ?

 Minor comments:

- line 170 , the font size needs correction

- The abbreviation “APOBEC” should be explained in the legend to Fig. 1

- Table 1 lacks explanation of some abbreviations

- line 209, abbreviations should be explained

Comments on the Quality of English Language

The English language needs some improvement. E.g. line 66. Instead of "...to improve schizophrenia" another term should be used: "improve the functioning of patients suffering from schizophrenia" or "alleviate the symptoms of schizophrenia".

Author Response

In this review, Jiaxiang Xie and colleagues analyzed distinctive patterns of 5-methylcytosine and 5-hy-droxymethylcytosine in schizophrenia. This analysis is justified by research indicating that the alteration in epigenetic pathway was associated with clinical features and brain dysfunctions in schizophrenia. Although the rationale seems quite reasonable, the manuscript has many serious shortcomings. I suggest the Authors clarify or provide more details regarding the following major and minor points.

Response: We thank the reviewer for the suggestions. We have addressed the reviewer’s concerns point-by-point as follows:

Major concerns:

  1. It is well known that many neuropsychiatric diseases, and others, are associated with epigenetic changes. Many post-mortem tests showed that that the level of DNMT and DNA methylation is higher in patients with schizophrenia than in healthy people. The cases of epigenetic modifications of the promoters of genes encoding COMT, Reelin and others described in this manuscript are well documented. They are not specific to schizophrenia and are also observed in other neuropsychiatric diseases. Therefore, the authors should clearly state the purpose of preparing this review.

Response: As the reviewer noted, these have been widely demonstrated that DNA methylation is changed in neurodevelopment, neurodegenerative and neuropsychiatric diseases. And DNMT higher expression is higher in schizophrenia patients. Abnormal COMT and Reelin indeed are popular in neuropsychiatric disorders. We have added the descriptions in lines 101-102, lines 118-120, lines 139-141, lines 144-146 and lines 161-162. Although these findings are similar, their contribution to the physiopathological progresses of schizophrenia cannot be denied. Our analysis of rhesus monkey brains unexpectedly revealed that cerebellum-specific 5hmC was concentrated in genes associated with schizophrenia. This has aroused great interest in us to summarize the role of DNA methylation and related enzymes in this disease. Our review summarized recent advances in 5mC and 5hmC changes in schizophrenia, especially newest studies from 2020 to 2022. Taken together, we tried to find out the clues of DNA methylation to schizophrenia and find new directions for epigenetic modification in schizophrenia.

  1. Did the authors intend to present a systematic review? Because if so, they should use the methodology applicable to reviews of this type.

Response: Strictly speaking, this review is not a systematic review. We focused on significant changes and novel findings in 5mC and 5hmC in schizophrenia. In particular, the dynamic 5hmC has not yet been summarized. So not all data may be included.

  1. It should be emphasized what is new in the approach presented here. What's new in the analysis of generally known data from the scientific literature on epigenetic changes described in schizophrenia? Any new conclusions?

Response: Currently, the classification of schizophrenia is not comprehensive. DNA methylation may be different in different types of schizophrenia, and we summarized the DNA methylation changes in several atypical schizophrenia classifications. Please see lines 286-305.

 Minor comments:

- line 170 , the font size needs correction

Response: The phrase "through the participation of APOBEC" has been corrected to match the text size.

- The abbreviation “APOBEC” should be explained in the legend to Fig. 1

Response: APOBEC is explained in detail in line 225, and we also explain it in the legend to Fig. 1

- Table 1 lacks explanation of some abbreviations

Response: The abbreviations in table 1 have been explained at their first appearance in the text.

- line 209, abbreviations should be explained

Response: The abbreviations have been explained.

Comments on the Quality of English Language

The English language needs some improvement. E.g. line 66. Instead of "...to improve schizophrenia" another term should be used: "improve the functioning of patients suffering from schizophrenia" or "alleviate the symptoms of schizophrenia".

Response: It has been replaced and the English in the text has been improved.

Round 2

Reviewer 2 Report

Comments and Suggestions for Authors

The authors have been able to correctly revise my previous comments on the manuscript. I have no further comments.